# Electrophoretic Determination of Trimethylamine (TMA) in Biological Samples as a Novel Potential Biomarker of Cardiovascular Diseases Methodological Approach

**DOI:** 10.3390/ijerph182312318

**Published:** 2021-11-23

**Authors:** Marek Konop, Mateusz Rybka, Emilia Waraksa, Anna K. Laskowska, Artur Nowiński, Tomasz Grzywacz, Wojciech J. Karwowski, Adrian Drapała, Ewa Maria Kłodzińska

**Affiliations:** 1Department of Experimental Physiology and Pathophysiology, Laboratory of Centre for Preclinical Research, Medical University of Warsaw, 02-106 Warsaw, Poland; mateuszrybka@mp.pl (M.R.); art.nowinski@gmail.com (A.N.); adrapala@wum.edu.pl (A.D.); 2Department of Analytical Chemistry and Instrumental Analysis, Institute of Sport—National Research Institute, 01-879 Warsaw, Poland; emilia.waraksa@insp.waw.pl; 3Department of Pharmaceutical Microbiology, Centre for Preclinical Research and Technology (CePT), Faculty of Pharmacy, Medical University of Warsaw, Banacha 1B, 02-097 Warsaw, Poland; anna.laskowska@wum.edu.pl; 4Department of Sport, Institute of Physical Culture, Kazimierz Wielki University, 85-064 Bydgoszcz, Poland; tomasz.grzywacz@insp.waw.pl; 5Department of Measurement and Electronics, Faculty of Electrical Engineering, Automatics, Computer Science and Biomedical Engineering, AGH University of Science and Technology, 02-106 Kraków, Poland; wojciech.karwowski@saintlazarus.pl

**Keywords:** cardiovascular diseases in athletes, biomarkers, determination of trimethylamine (TMA), capillary electrophoresis (CE)

## Abstract

In competitive athletes, the differential diagnosis between nonpathological changes in cardiac morphology associated with training (commonly referred to as “athlete’s heart”) and certain cardiac diseases with the potential for sudden death is an important and not uncommon clinical problem. The use of noninvasive, fast, and cheap analytical techniques can help in making diagnostic differentiation and planning subsequent clinical strategies. Recent studies have demonstrated the role of gut microbiota and their metabolites in the onset and the development of cardiovascular diseases. Trimethylamine (TMA), a gut bacteria metabolite consisting of carnitine and choline, has recently emerged as a potentially toxic molecule to the circulatory system. The present work aims to develop a simple and cost-effective capillary electrophoresis-based method for the determination of TMA in biological samples. Analytical characteristics of the proposed method were evaluated through the study of its linearity (R^2^ > 0.9950) and the limit of detection and quantification (LOD = 1.2 µg/mL; LOQ = 3.6 µg/mL). The method shows great potential in high-throughput screening applications for TMA analysis in biological samples as a novel potential biomarker of cardiovascular diseases. The proposed electrophoretic method for the determination of TMA in biological samples from patients with cardiac disease is now in progress.

## 1. Introduction

Athletes are seen to be healthy, physically fit, and able to tolerate extremes of physical endurance. It seems improbable that such athletes may have underlying life-threatening, cardiovascular abnormalities. Regular physical activity promulgates cardiovascular fitness and lowers the risk of cardiac disease. However, with intense physical exertion and the harboring of an underlying disease, athletes may suffer sudden cardiac death. The echocardiographic examinations in athletes showed left ventricular hypertrophy of the heart muscle. Knowledge of those changes may help identify cardiovascular abnormalities that can cause sudden death from the heart known as an “athlete’s heart” [1].

Trimethylamine is an organic compound with the formula C_3_H_9_N or (CH_3_)_3_N. It appears as a colorless gas with a fishlike odor at low concentrations that changes to an ammonialike odor at higher concentrations. In the human body, trimethylamine (TMA) is exclusively a microbiota-generated product of nutrients (lecithin, choline, L-carnitine) from a normal diet, from which two diseases seem to originate, trimethylaminuria (or Fish Odor Syndrome) and cardiovascular disease through the proatherogenic property of its oxidized liver-derived form [2,3].

In humans, the ingestion of certain plants or animals can elevate trimethylamine levels in the body. Food containing choline, lecithin, and L-carnitine delivers colon microbiota, a substrate to synthesize TMA. These compounds can be found in various products in people’s diets all over the world. For example, peanuts, liver, and eggs are especially rich in choline [4]. However, taking into consideration the role of every product in the American diet, data has shown that the highest choline levels are provided by eating poultry, meat, dairy foods, rice, fish, pasta, and egg-based dishes [5]. The second one, lecithin is found in many different products including red meat, seafood, eggs, organ meat, cooked green vegetables, kidney beans, soybeans, and black beans. While food of animal origin such as beef steak, milk, and cheese are the best natural sources of L-carnitine, wheat products deliver one hundred times less L-carnitine than meat, although it can still be provided by whole wheat bread [6]. However, the bioavailability of L-carnitine from food varies depending on diet composition. Those who are adapted to high carnitine diets have lower L-carnitine bioavailability. Considering the role of these compounds in trimethylamine synthesis, it is important to note that products such as meat deliver more than one substrate for TMA production [7,8].

Three microbial pathways of TMA colon production have been described till now. Based on the similarities of ethanolamine and TMA, the *Desulfovibrio genus* was identified as responsible for the anaerobic degradation of choline [9]. Choline–TMA–lyase activity encoded by part of a gene cluster was confirmed with a genetic knockout method and comparing choline-degrading organisms and non-choline-degrading strains. A second microbial pathway generates TMA through L-carnitine hydroxylation. It has been shown that this pathway is active in *Serratia* [10] and *Acinetobacter* [11]. Carnitine monooxygenase is the crucial enzyme in L-carnitine conversion. It is encoded by two genes, CntA and CntB, and it was identified as a Rieske type oxygenase/reductase. The last known pathway was described by Koeth et al. [12]. The researchers identified γ-butyrobetaine (γ-BB) as the crucial intermediate of L-carnitine of gut microbial catabolism in mice. The processes of degrading L-carnitine to γ-BB commence in the mouse ileum, however, the TMA production from γ-BB is restricted to the colon, including the cecum [12].

Despite the fact that only a minimal effort has been made to assess the microbiota background of TMA production, Koeth et al. examined the colon microbiome of 10 mice fed standard and 11 mice fed L-carnitine-supplemented diet chow. The composition of the microbiota in mice fed the L-carnitine supplemented diet had a significantly increased number of *Anaeroplasma*, *Prevotella*, and *Mucispirillum*. Following up on the received data and the observation which showed that vegetarians and vegans are likely to produce less TMAO, the team decided to analyze the microbiota composition in 23 vegans. The study demonstrated that in the feces of vegans there is an increased amount of *Lachnospira* spp., which anticorrelates with the TMAO plasma levels regardless of diet [13].

Several different methods have been developed to improve the analysis of TMA, including GC or GC–MS, using a headspace technique for sample pretreatment [14,15,16], TMA-selective electrodes [17,18], enzymatic methods [19], spectrophotometric methods [20,21], the NMR method [22], ion chromatography [22], and high-performance liquid chromatography (HPLC) [23,24]. An alternative analytical technique is capillary electrophoresis (CE) [25,26].

Capillary electrophoresis is an analytical technique that separates ions based on their electrophoretic mobility with the use of an applied voltage. The electrophoretic mobility is dependent upon the charge of the molecule, the viscosity, and the atom’s radius [27]. The rate at which the particle moves is directly proportional to the applied electric field—the greater the field strength, the faster the mobility. Neutral species are not affected, only ions move with the electric field. If two ions are the same size, the one with the greater charge will move the fastest. For ions of the same charge, the smaller particle has less friction and an overall faster migration rate [28]. Capillary electrophoresis is used most predominately because it gives faster results and provides high-resolution separation. It is a useful technique because there is a large range of detection methods available. It should be mentioned that TMA is difficult to determine by conventional HPLC and CE because it does not absorb strongly in the UV-visible region. Since TMA is rather insensitive towards common UV detectors, a complex chemical derivatization is generally required [26]. Capillary electrophoresis (CE) offers fast and high-resolution separation of the charged analytes from small injection volumes, is cost-effective, generates minimal waste, and can separate complex mixtures of analytes [29,30]. Moreover, further selectivity may be achieved in CE by employing pseudo stationary phases (solution-based additives present in the separation buffer, which effect the separation of analytes based on their differential associations). The use of pseudo stationary phases rather than true stationary phases in the CE-based methods reduces problems with irreproducibility between capillaries, and, furthermore, it is simpler than introducing selectivity via the more time-consuming process of the immobilization of nanomaterials to form inner capillary wall coatings [31,32,33]. Screening gut microbial trimethylamine production by fast and cost-effective capillary electrophoresis was also utilized by García-Cañas et al. [34].

The present study aimed to develop and validate simple, fast, low-cost, and efficient CE-based methods with the highest potential for laboratory applications for the determination of TMA in biological samples as a potential biomarker of early heart disease. The assay achieved satisfactory validation parameters, such as linearity (R^2^ > 0.9950), the limit of detection (LOD = 1.2 µg/mL), and the limit of quantification (LOQ = 3.6 µg/mL). Therefore, the proposed method can be successfully applied to the routine analysis of real samples in laboratories equipped with CE.

## 2. Materials and Methods

### 2.1. Chemicals and Standards

The trimethylamine solution (45 wt. % in H_2_O), sodium hydroxide, 2,4′-dibromoacetophenone (DBA), and acetonitrile were purchased from Sigma-Aldrich (Poznan, Poland). Sodium hydroxide was purchased from Avantor Performance Materials Poland S.A. (Gliwice, Poland).

### 2.2. Haemolysis Assay

The hemolytic activity of compounds (TMA/TMAO) was evaluated according to Mazzarino et al. [35] with modifications. Briefly, human and rat red blood cells (RBC) were obtained from a healthy volunteer/animal. Samples were centrifuged at 2500 rpm for 10 min, and plasma was removed. Samples were washed 3 times and resuspended in a sterile PBS. RBC samples were diluted in a sterile PBS according to donor/animal haematocrit to obtain a 10% and 2% RBC suspension. Two percent haematocrit was incubated with serial concentrations of compounds (1:1) at 37 °C for 60 min. Next, samples were centrifuged at 4500 rpm for 5 min, and 100 µL of the supernatant from each sample was transferred to a 96-well plate. Absorbance was measured at 540 nm. A value of 100% hemolysis was determined by incubation of a 10% RBC suspension in distilled water (1:9). For the negative control (0% hemolysis), a 2% RBC suspension was incubated with a PBS (1:1). The hemolysis rate was calculated according to the following equation:Haemolysis [%] = (A − A0%)/(A100% − A0%) × 100%
where A–absorbance of the sample incubated with compound, A100%–absorbance of reference (100% hemolysis), A0%–absorbance of negative control (0% hemolysis).

### 2.3. Cell Culture

Rat fibroblasts Rat2 (ATCC-CRL-1764) were cultured in Dulbecco’s Modified Eagle Medium (DMEM) with a 10% (*v*/*v*) heat-inactivated fetal bovine serum (FBS), and a 1% (*v*/*v*) penicillin–streptomycin solution. Cells were maintained at 37 °C and 5% CO2. Passages 3 to 6 were used in all experiments.

### 2.4. Cell Viability Assay

A Cell Proliferation Kit I (MTT) (Roche) was used to evaluate cell viability. Cells were seeded in 96-well plates at a density of 3 × 10^3^ cells/well and incubated for 24 h. Next, compounds (TMA/TMAO) (0–12.5 mg/mL) were added to the cells, and plates were incubated at 37 °C. After 24 h and 48 h, 10 µL of an MTT solution was added and incubated for 4 h at 37 °C. Next, 100 µL of a SDS solution was added, and plates were incubated overnight at 37 °C. Absorbance was measured at 570 and 690 nm using a microplate reader. The experiment was performed in triplicate.

### 2.5. Samples

In this study, we evaluated urine, feces, and serum samples from healthy and spontaneously hypertensive rats. The study’s experimental design was approved by the 2nd Local Ethics Committee for Experiments on Animals at the Warsaw University of Life Sciences, Warsaw, Poland (Certificate of approval No. WAW2/082/2018). The stock solutions of standards were prepared in ultrapure water or acetonitrile.

### 2.6. Sample Preparation

The derivatization process described by García-Cañas et al. [34] was applied in our study with slight modifications. The reaction between DBA and TMA was performed to obtain the maximum yield of the product in the shortest reaction time. Rat plasma, urine, and fecal supernatant samples were used. Under optimized conditions, TMA was derivatized using a DBA reagent. A total of 100 μL of the sample (or standards) was mixed with 300 μL of 80 mM DBA in acetonitrile in a 2 mL screw-capped tube. The reaction mixture was placed in a dry block heater at 70 °C for 60 min. After 1 h, the reaction was stopped by cooling down to 4 °C. Next, the sample was evaporated to dryness under a nitrogen stream at room temperature. Then, 300 μL of distillate water was added and vortexed for 5 min and centrifugated at 15,000× *g* for 5 min at room temperature. Finally, the supernatants were directly injected into the CE-UV system.

### 2.7. Instrumentation: Capillary Zone Electrophoresis

A CE analysis was carried out using an Agilent 7100 series CE instrument (Agilent Technologies, Santa Clara, CA, USA) equipped with a UV-VIS diode array detector. A fused-silica capillary (365 µm outer diameter and 50 µm internal diameter) was purchased from Composite Metal Services (Worcester, UK). The capillary effective length was 56 cm. The temperature of both the capillary and samples was maintained at 20 °C. A borate buffer (20 mM; pH 9.3) was used as a running buffer. Detection was performed at λ = 210, 214, and 254 nm. A constant voltage of 20 kV (maximal current did not exceed 50 µA) was used for all of the measurements. Samples were pressure-injected at 50 mbar for 10 s. To provide high reproducibility of the migration times and to avoid solute adsorption, the capillary was washed between injections as follows: 0.1 M NaOH for 1 min, followed by water for 1 min, and then equilibrated with the running buffer. The pressure used for rising of the capillary was 900 mbar.

## 3. Results

### 3.1. Haemolysis Assay

Rat and human RBCs were similarly susceptible to the exposure to tested compounds (TMA/TMAO). The TMA-induced hemolysis was concentration dependent. In the case of rat RBCs, a significant increase in the level of the hemolysis was observed at a concentration of >0.196 mg/mL. For human RBCs, an increased hemolysis was observed at a concentration of >0.391 mg/mL (Figure 1). Above a concentration of 0.196 mg/mL, the difference between TMA-induced hemolysis in human and rat samples was statistically significant. The TMAO-induced hemolysis was not concentration dependent. For rat and human RBCs, the level of hemolysis was similar to the hemolysis of RBCs incubated with a PBS (Figure 1).

### 3.2. Cell Viability Assay

It was observed that the effect of TMA and TMAO on rat fibroblasts was time dependent (Figure 2). After 24 h, the viability of TMA-treated cells and control cells was comparable. After 48 h, the viability of cells treated with TMA (<6.25 mg/mL) was 10% lower than the viability of untreated cells. At concentrations of >6.25 mg/mL, the decrease in cell viability was higher. Cells were susceptible to exposure to TMAO. In the case of 24 h incubation, a 5–10% decrease in cell viability was observed but was not concentration dependent. After 48 h, fibroblasts viability decreased with an increasing concentration of TMAO. The viability of TMAO-treated cells was 10–25% lower than the viability of untreated fibroblasts.

### 3.3. Optimization of CE Conditions

A series of different buffer solutions with different concentrations and pH values were tested. The influence of the pH, conductivity, and the composition of the buffer solutions on the obtained electrophoretic results is shown in Figure 3.

The best separation was observed in the borate buffer at a concentration of c = 20 mM and pH = 9.3. The signals on the electropherograms were symmetrical, there were no changes in the migration times, and the current lines were observed. In other cases, disturbances in the baseline were observed, and the obtained current value was too high and unstable.

To improve the limit of detection of the proposed method, derivatization using a DBA reagent in biological samples was performed. Electropherograms obtained for TMA in different biological matrices with and without the addition of the IS are shown in Figure 4.

### 3.4. Method Validation

The study was focused on creating fast, reliable, repeatable, and environmentally-friendly methods utilizing minimal sample preparation, which can be utilized in a routine analysis of TMA as a potential biomarker of cardiovascular diseases. The method linearity was determined as the coefficient of determination (R2). The linearity range was evaluated within 6.1–3125.0 µg/mL. The nine-point calibration curve was constructed by plotting peak areas by the concentrations. The calibration curve was proven to be linear (R2 > 0.9950) in the investigated concentration range (6.1–3125.0 µg/mL). The LOD and LOQ were evaluated based on the calibration curve in the concentration range of 6.1–3125.0 µg/mL. The LOD and LOQ were calculated as follows: LOD = 3.3 × Sb/a and LOQ = 10 × Sb/a, where Sb is the standard deviation of the incept of the calibration curve, and a is the slope of the calibration curve. The LOD (LOQ) was 1.2 µg/mL (3.6 µg/mL) for the proposed analytical method.

The performance of the analytical method was evaluated in terms of linearity, the limit of detection (LOD), and the limit of quantitation (LOQ). The obtained results are presented in Table 1.

### 3.5. Applicability of the Method

To evaluate the applicability of the method, biological samples such as urine, plasma, and fecal supernatant from rats and athletes were analyzed. The obtained results are shown in Table 2.

In individual cases of the analyzed biological samples taken from healthy volunteers, an increase in the TMA concentration was observed (Figure 4h). However, studies on a larger group of people (e.g., athletes) representing various sports disciplines are needed.

## 4. Discussion

Lately, microbiota composition is being considered an important factor in CVD development [36]. The consumption of TMA precursors such as choline and betaine were identified as the risk factors in CVD. Six prospective cohort studies, performed in three different countries, showed increased CVD incidence in five of them and increased CVD mortality in two [37,38,39,40,41,42,43]. Therefore, colon TMA production can be targeted in choline-linked cardiovascular diseases. Despite the fact that most of the studies were focused on elevated TMAO levels in cardiovascular patients, there is evidence showing that higher TMA plasma levels can deliver more accurate information about CVD [44,45]. Besides, TMA was found to be cytotoxic for cardiomyocytes, likely due to the effects of LDH and other protein structures [1].

Capillary electrophoresis (CE) is a more sophisticated technique for the fast analysis of polar metabolites in a variety of biological matrices by only injecting a small volume (1–50 nL) from a few microliters of the sample into the vial. The application of CE for the determination of TMA in blood, feces, and urine samples has not been extensively investigated so far. One of the main difficulties regarding the analysis of TMA lies in its lack of fluorescence or UV absorption. For this reason, the derivatization of the sample before electrophoretic analysis with the use of a diode array detector that enables the simultaneous registration of signals spectra at different wavelengths in the UV-VIS region (190–600 nm) was proposed. The developed CE method was optimized to obtain the shortest analysis time while maintaining good resolution and sensitivity. The choice of the separation buffer was critical for obtaining a successful separation of analytes by CE. The first specific objective was aimed at finding a suitable electrolyte for the analysis of the TMA derivative, which is characterized by a low value of current, a stable baseline, and an electroosmotic flow. The borate buffer used for the analyses met these criteria. The applied voltage was too high when a 0.75 M formic acid at pH = 2.05 as a background electrolyte was used (Figure 3h). The high voltage reduces the analysis time, however, lower values enhanced separation. In the case of most of the tested buffers in the applied voltage above 15 kV, the observed current was too high (more than 100 µA) and caused Joule heating in the fused silica capillary. To obtain lower levels of the limit of detection, the injection time was also evaluated. Increasing the injection time decreased resolution, and broad signals were observed on the electropherograms. Therefore, a short injection time and high injection pressure were used. The optimal separation conditions were established at a 20 mM boric buffer at pH = 9.3 with an applied separation voltage of +20 kV. Under these conditions, it was possible to use the same running buffer during more than 20 analyses with no noticeable drift in migration time. TMA in biological samples (urine, blood, and feces) was detected in less than one minute. The total analysis time, including capillary preconditioning, was about three minutes.

The electrophoretic technique has several advantages over commonly-used chromatographic methods, including simplicity, low cost, versatility, low consumption of reagents, and a short analysis time. In the biological sample, the concentration level of TMA is usually very low. Therefore, an ideal solution would be the use of capillary electrophoresis coupled to mass spectrometry (CE–MS). The CE–MS allows to significantly improve the sensitivity, however, the use of this system requires strict conditions for the analysis, such as the use of appropriate buffers that are mutually exclusive in the case of a TMA analysis. Moreover, equipping the laboratory with a CE–MS is expensive and requires specialized personnel.

In the literature, there are only a few reports on the determination of TMA by CE. The proposed study is one of the first applications of electromigration techniques on the determination of TMA as a potential biomarker in cardiovascular diseases. The developed method can be used as a rapid screening test for identifying the risk of cardiovascular disease.

## 5. Conclusions

A simple CE method for the analysis of TMA in biological samples has been developed. The method proved to be rapid, simple, inexpensive, reproducible, and applicable to the analysis of TMA in urine, blood, and feces samples. The method was successfully applied to the analysis of rat urine and plasma as well as biological samples derived from athletes. The LOD (1.2 µg/mL) and LOQ (3.6 µg/mL) fully comply with the requirements for high-throughput screening applications. In the case of athletes, it is very important to detect potential biomarkers such as TMA that could indicate the development of early heart disease. A good solution seems to be a comparison of endurance, echocardiological, and electrophoretic tests. The proposed study provides a comprehensive look at the electrophoretic method and indicates its possible applicability as a biomarker in cardiovascular disease. Nevertheless, further studies involving a group of high-performance athletes and healthy volunteers are needed to demonstrate changes in the TMA concentration in biological samples.

## Figures and Tables

**Figure 1 ijerph-18-12318-f001:**
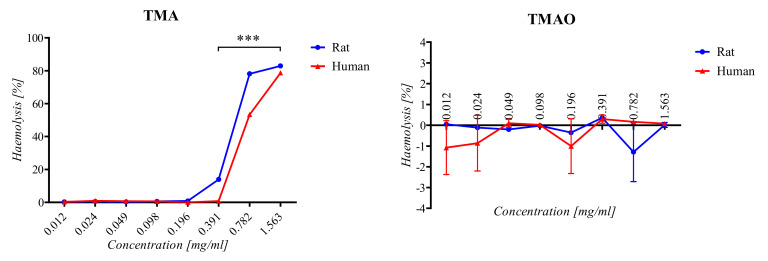
Hemolysis of rat and human red blood cells induced by TMA and TMAO. (*p*-value < 0.05 was considered significant *** *p* < 0.001, two-way analysis of variance, followed by Bonferroni post hoc tests)

**Figure 2 ijerph-18-12318-f002:**
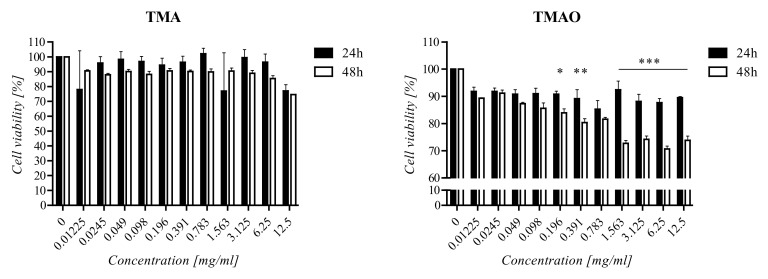
Viability of rat fibroblasts treated with TMA and TMAO for 24 and 48 h. The data are statistically significant when *p* < 0.05 using one-way analysis of variance. (* *p* < 0.05, ** *p* < 0.01, *** *p* < 0.001)

**Figure 3 ijerph-18-12318-f003:**
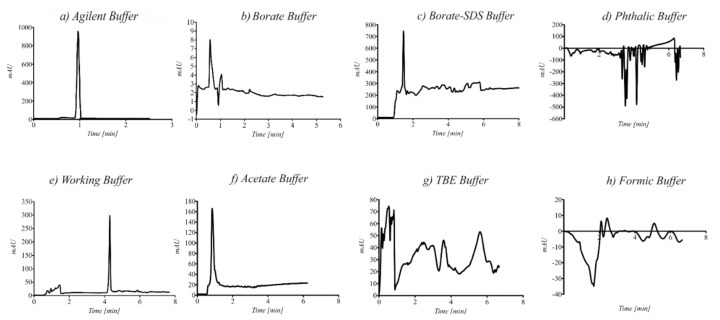
Electropherogram obtained for different buffer solutions before derivatization: (**a**) Agilent Buffer, (**b**) Borate Buffer, (**c**) Borate-SDS Buffer, (**d**) Phtalic Buffer, (**e**) Working Buffer, (**f**) Acetate Buffer, (**g**) TBE Buffer, (**h**) Formic Buffer.

**Figure 4 ijerph-18-12318-f004:**
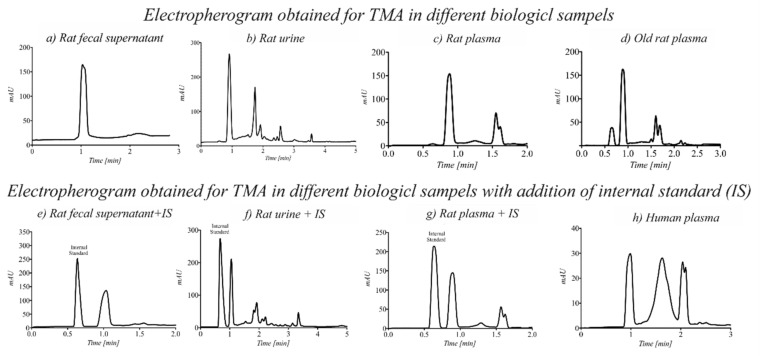
The electropherogram obtained for TMA was tested in different biological samples after the derivatization process. ((**a**) Rat fecal supernatant, (**b**) Rat urine, (**c**) Rat plasma, (**d**) Old rat plasma, (**e**) Rat fecal supernatant + Internal Standard, (**f**) Rat urine + Internal Standard, (**g**) Rat plasma + Internal Standard, (**h**) Human plasma)

**Table 1 ijerph-18-12318-t001:** Method validation data.

Characterization Parameter	TMA [µg/mL]
Calibration range	6.1–3125.0
Calibration curve equation	y = 0.0118x + 9.5357
Linearity (R^2^)	0.9950
LOD	1.2
LOQ	3.6

R^2^: coefficient of determination; LOD: limit of detection; LOQ: limit of quantification.

**Table 2 ijerph-18-12318-t002:** Information on the results obtained from the analysis of the real samples.

Real Urine Samples		Real Plasma Samples		Real Faces Samples	
Sample Name	Average TMA Concentration ± U [µg/mL] *	Sample Name	Average TMA Concentration ± U [µg/mL] *	Sample Name	Average TMA Concentration ± U [µg/mL] *
U1	1.0 × 10^2^ ± 1.7	P1	3.4 × 10 ± 3.6	F1	266.6 ± 3.2
U2	6.8 × 10 ± 1.0	P2	nd **	F2	429.9 ± 3.2
U3	9.0 × 10 ± 8.3	P3	nd **	F3	nd **
U4	2.9 × 10 ± 3.5	P4	1.25 × 10^2^ ± 0.19	F4	562.1 ± 6.7
U5	1.1 × 10^3^ ± 3.6 × 10	P5	nd **	F5	61.7 ± 1.8
U6	1.3 × 10^2^ ± 1.0 × 10	P6	2.1 × 10 ± 3.2	F6	28.6 ± 1.8
U7	3.1 × 10^2^ ± 4.1 × 10	P7	7.8 × 10 ± 5.3	F7	nd **
U8	1.2 × 10^2^ ± 1.0 × 10	P8	nd **	F8	578.6 ± 1.1
U9	7.9 × 10 ± 1.2 × 10	P9	nd **	F9	31.1 ± 1.8
U10	3.5 × 10^2^ ± 2.8 × 10	P10	nd **	F10	71.4 ± 2.1
U11	3.5 × 10^2^ ± 3.1 × 10	P11	nd **	F11	244.0 ± 9.6
U12	4.0 × 10^2^ ± 4.4 × 10	P12	nd **	F12	277.6 ± 3.2
U13	1.6 × 10^2^ ± 2.4 × 10	P13	nd **	F13	nd **
U14	1.5 × 10^2^ ± 2.1 × 10	P14	nd **	F14	522.5 ± 27.9
U15	2.6 × 10^2^ ± 1.1 × 10	P15	1.1 × 10^3^ ± 1.1 × 10^2^	F15	102.3 ± 1.1

* expressed as an average concentration (*n* = 2) ± U, where U is the expanded uncertainty using a coverage factor of 2, which gives a level of confidence of approximately 95%, ** nd—<LOD.

## Data Availability

Not applicable.

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
