# Peer review of "Electrophoretic Determination of Trimethylamine (TMA) in Biological Samples as a Novel Potential Biomarker of Cardiovascular Diseases Methodological Approach"

_ijerph, 2021, doi:10.3390/ijerph182312318_

Round 1

Reviewer 1 Report

The manuscript entitled Electrophoretic determination of trimethylamine (TMA) in biological samples as a novel potential biomarker of cardiovascular diseases methodological approach could have a scientific impact, but some correction must be made.

I recommend this work for publication with minor corrections:

Line 30 – In sentence …. nutrients into trimethylamine (TMA) trimethylamine-N-oxide (TMAO)…after (TMA)and (TMAO) authors should put a comma.

Please do not repeat abbreviations in the abstract.

Fig.2 – The authors should also provide an explanation for the time in the case of TMAO

Fig. 3 – Better resolution is required and also the authors have Figure 3 twice.  

Line 228 - Please correct R2

Line 229 - within6.1 there should be a space between word and number

Line 231 - Please correct  (R2>0.9950)

Line 234 – Please correct “incept“

Authors should check the value of LOD. It is not identical in the text and in Table 1. Also explain that „LOD and LOQ were evaluated based on the calibration curve in the concentration range of 103.2 - 52867.5 μmol/L“

Line 251 - Fig 2e is not included in the manuscript

Line 279 – Fig. 1h is not included in the manuscript

The author should include section of statistical analyses that are used.

Carefully correct the reference section.

Author Response

Line 30 – In sentence …. nutrients into trimethylamine (TMA) trimethylamine-N-oxide (TMAO)…after (TMA)and (TMAO) authors should put a comma. 

We are grateful for underlining this mistake. The Abstract was changed.  

Please do not repeat abbreviations in the abstract. 

We are grateful for underlining this mistake. The Abstract was changed.  

Fig.2 – The authors should also provide an explanation for the time in the case of TMAO 

Fig. 3 – Better resolution is required and also the authors have Figure 3 twice.   

Line 228 - Please correct R2 

We are grateful for underlining this mistake. It was corrected. 

Line 229 - within6.1 there should be a space between word and number 

We are grateful for underlining this mistake. It was corrected. 

Line 231 - Please correct  (R2>0.9950) 

We are grateful for underlining this mistake. It was corrected. 

Line 234 – Please correct “incept“ 

We are grateful for underlining this mistake. It was corrected. 

Authors should check the value of LOD. It is not identical in the text and in Table 1. Also explain that „LOD and LOQ were evaluated based on the calibration curve in the concentration range of 103.2 - 52867.5 μmol/L“ 

We are grateful for underlining this mistake. It was a typographical error. It was corrected. The sentence "The LOD and LOQ were evaluated based on the calibration curve in the concentration range of 103.2-52867.5ug/mL" was changed to "The LOD and LOQ were evaluated based on the calibration curve in the concentration range of 6.1 – 3125.0 µg/mL. Mistake above resulted from the primary calculations in µmol/L. 

Line 251 - Fig 2e is not included in the manuscript 

 We are grateful for underlining this mistake. It was a typographical error and it was corrected for appropriate figure 4 h.  

Line 279 – Fig. 1h is not included in the manuscript 

We are grateful for underlining this mistake. It was a typographical error and Fig. 1h was corer for appropriate fig. 3h 

The author should include section of statistical analyses that are used. 

Information about statistical analyses were added into description below fig. 2. and highlighted.  

Reviewer 2 Report

in general the manuscript is presented correctly,

a lot of bibliography should be added to mention in introduction and more in discussion of results.

I would like you to add more references of similar cases, so that there can be a clear difference of what the impact of this study is compared to those that have already been done.

Author Response

(The authors gave the same response as above.)

Reviewer 3 Report

Konop et al. have investigated into development of low cost, simple, fast and efficient electrophoretic method for determination of trimethylamine (TMA) in biological samples in early heart disease conditions using TMA as a potential biomarker. They successfully applied this method for rat urine, plasma as well as biological samples for the athletes. They found 1.2 ug/mL as limit of detection and 3.6 ug/mL as limit of quantification using this method.

The manuscript can not be accepted for this journal. First, the manuscript is very poorly written. The authors have described "why" this method is useful in athletes but did not cover "why" this method is warranted over other methods available. They should cover enough background in introduction on current methods/techniques that are available for the determination of TMA in samples.

The authors should add why the haemolysis assay (Figure-1) and cell viability assay (Figure-2) was performed. What were the implications of performing this experiments?

The authors should also add background about this technique in the introduction. There is plenty of information about microbiota and different sources of TMA and TMAO, but if the paper is about the method then it should be described properly in the manuscript.

Author Response

Reviver 3:  

Konop et al. have investigated into development of low cost, simple, fast and efficient electrophoretic method for determination of trimethylamine (TMA) in biological samples in early heart disease conditions using TMA as a potential biomarker. They successfully applied this method for rat urine, plasma as well as biological samples for the athletes. They found 1.2 ug/mL as limit of detection and 3.6 ug/mL as limit of quantification using this method. 

  1. The manuscript can not be accepted for this journal. First, the manuscript is very poorly written. The authors have described "why" this method is useful in athletes but did not cover "why" this method is warranted over other methods available. They should cover enough background in introduction on current methods/techniques that are available for the determination of TMA in samples.

Dear Reviver, thank you very much fort this comment. Several information’s were added into introduction and highlighted.  

  1. The authors should add why the haemolysis assay (Figure-1) and cell viability assay (Figure-2) was performed. What were the implications of performing this experiments?

Thank you very much for this question. Both tests were performed to determine the biological effect of TMA and TMAO which may be detrimental to the circulatory system. To the best of our knowledge this is the  one of the first studies that showed that TMA, but not TMAO can cause haemolysis in a concentration-dependent manner.   

This data can correlate with cell viability assay. Jaworska et al. [1] found that TMA but not TMAO decreased the proliferation and viability of cardiomyocytes. Moreover, the concomitant treatment with TMAO protected cardiomyocytes against the cytotoxic effect of TMA. The protective action of TMAO was evident after 10 days of treatment. 

  1. The authors should also add background about this technique in the introduction. There is plenty of information about microbiota and different sources of TMA and TMAO, but if the paper is about the method then it should be described properly in the manuscript.

Dear Reviver, thank you very much for this comment. Several pieces of information were added into an introduction and highlighted.  

  1. Jaworska, K.; Hering, D.; Mosieniak, G.; Bielak-Zmijewska, A.; Pilz, M.; Konwerski, M.; Gasecka, A.; Kaplon-Cieślicka, A.; Filipiak, K.; Sikora, E.; et al. TMA, a forgotten uremic toxin, but not TMAO, is involved in cardiovascular pathology. Toxins (Basel). 2019, doi:10.3390/toxins11090490. 

Round 2

Reviewer 3 Report

The authors have addressed the concerns raised in the earlier report. It has all the details required in the introduction, thus the manuscript can be accepted in its current form.